# Aptamers Targeting Membrane Proteins for Sensor and Diagnostic Applications

**DOI:** 10.3390/molecules28093728

**Published:** 2023-04-26

**Authors:** Nilufer Kara, Nooraldeen Ayoub, Huseyin Ilgu, Dimitrios Fotiadis, Muslum Ilgu

**Affiliations:** 1Department of Biological Sciences, Middle East Technical University, Ankara 06800, Turkey; 2Institute of Biochemistry and Molecular Medicine, University of Bern, CH-3012 Bern, Switzerland; 3Roy J. Carver Department of Biochemistry, Biophysics and Molecular Biology, Iowa State University, Ames, IA 50011, USA; 4Department of Veterinary Microbiology and Preventive Medicine, Iowa State University, Ames, IA 50011, USA; 5Aptalogic Inc., Ames, IA 50014, USA

**Keywords:** aptamers, aptasensor, biosensor, diagnostics, membrane proteins, SELEX

## Abstract

Many biological processes (physiological or pathological) are relevant to membrane proteins (MPs), which account for almost 30% of the total of human proteins. As such, MPs can serve as predictive molecular biomarkers for disease diagnosis and prognosis. Indeed, cell surface MPs are an important class of attractive targets of the currently prescribed therapeutic drugs and diagnostic molecules used in disease detection. The oligonucleotides known as aptamers can be selected against a particular target with high affinity and selectivity by iterative rounds of in vitro library evolution, known as *Systematic Evolution of Ligands by EXponential Enrichment* (SELEX). As an alternative to antibodies, aptamers offer unique features like thermal stability, low-cost, reuse, ease of chemical modification, and compatibility with various detection techniques. Particularly, immobilized-aptamer sensing platforms have been under investigation for diagnostics and have demonstrated significant value compared to other analytical techniques. These “aptasensors” can be classified into several types based on their working principle, which are commonly electrochemical, optical, or mass-sensitive. In this review, we review the studies on aptamer-based MP-sensing technologies for diagnostic applications and have included new methodological variations undertaken in recent years.

## 1. Introduction

The biological membranes are essential for cellular life. They form and organize cells’ defining boundaries, e.g., by separating the interior from the outside environment. Most cell membranes consist of about half lipid and half protein by weight. Membrane proteins (MPs) are proteins embedded or attached to biological membranes and are thus classified as peripheral or integral. Integral MPs fully span the lipid bilayer, while peripheral MPs partially associate with the lipid bilayer or with integral MPs (Figure 1). MPs are broadly amphipathic, having hydrophobic and hydrophilic regions, and they distribute asymmetrically through the membrane, with some being modified with carbohydrate moieties [1]. The hydrophobic nature of the lipid bilayer core limits the possible transmembrane protein structures to α-helix and β-barrel structures. MPs perform most of the specific functions of membranes. For example, receptors and transporters play critical roles in transmitting information and molecules into a cell or organelle. Cell receptors sense external cues and integrate them to respond through coordinated signal transduction pathways. Considering their important physiological roles, MPs are crucial in medicine, pharmacology, and drug discovery representing the vast majority of therapeutic drug targets [2]. These clinical drug targets include channels, transporters, and in particular, G protein-coupled receptors (GPCRs) [3].

Functional characteristics of organelles and cells are determined by the protein compositions at different physiological states [4]. Changes in genetic composition or level of MPs are associated with various diseases, for which rapid and early identification become possible with the development of specific recognition elements for these MPs [5]. In the last three decades, nucleic acid aptamers have been selected and optimized for numerous MPs, and they have become antibody alternatives to be integrated into diagnostic tools. Here, we summarized “aptasensors” for MPs developed based on electrochemical, optical, or mass-sensitive technologies. As MPs serve as disease biomarkers and key virulence factors of bacterial and viral pathogens, we believe that advancements in aptamer-based diagnostic tools with novel methodological variations will help early diagnosis of diseases and future events like the COVID-19 pandemic or other unexpected outbreaks.

## 2. Membrane Proteins: Role in Diseases and Potential as Biomarkers

Approximately 30% of the total human proteome is composed of MPs [4,6]. Having important physiological roles, MPs are critical in the development and progress of various pathological conditions. For example, G-protein-coupled receptors (GPCRs) are a well-characterized class of membrane receptors encoded by more than 800 human genes [7]. They are dynamic signaling receptors that play roles in major signaling pathways, such as those related to the actions of drugs, toxins, hormones, and neurotransmitters, sensing light and odors, and regulating water reabsorption and blood calcium levels [8,9]. Acquired and inherited genetic mutations result in GPCR dysfunctions and, thus, disorders like retinitis pigmentosa, hypo- and hyperthyroidism, nephrogenic diabetes insipidus, fertility disorders, and different carcinomas [10]. Channelopathies, such as long QT syndrome and cystic fibrosis, are another group of disease conditions that can arise from defects in the class of transmembrane proteins comprising ligand- and voltage-activated ion channels [11,12]. These channels are known to regulate ion and water balance, membrane potentials, and signal transduction.

Receptor tyrosine kinases (RTKs) involve a class of membrane receptors encoded by 60 human genes and participate in important functions, such as regulation of cell survival, metabolism, proliferation, and differentiation [13]. Dysfunctional RTKs result in developmental problems leading to diseases, including diabetes, atherosclerosis, and cancer. For example, RTKs are well-studied and targeted therapeutics for cancer [14] and diabetes [15].

Transporter proteins are another important class of MPs encoded by 10% of human genes [16,17]. They include families of active, ATP-dependent transporters known as ATPases that are responsible for cell survival by achieving different ionic equilibria of sodium, potassium, calcium, and H^+^ ions (i.e., P-type ATPases) [18,19]. Malfunction in these transporters leads to various diseases ranging from migraines, heritable deafness, and balance disorder to renal diseases, copper-related disorders, and cancers [20,21]. Transporters are emerging as attractive drug targets [22]. Solute carrier (SLC) proteins are a rich and diverse group of transporters that facilitate the transport of various molecules, including glucose, amino acids, fatty acids, urea, bile salts, large organic ions, nucleosides, and neurotransmitters [17]. Defects in SLC transporters, therefore, have implications in neurodegenerative diseases [23] and many metabolic disorders [17].

ATP-binding cassette (ABC) proteins include a group of transporters with various unique functions like the transport of peptides, phospholipids, bile materials (e.g., salts, cholesterol, etc.), and surfactants, and the presentation of antigens [4,24]. Multidrug resistance (MDR)-ABC proteins are involved in the metabolism and transport of many foreign materials, including endo- and xenobiotics, anticancer drugs, and partially detoxified drug metabolites. Alterations in the structure and expression of MDR-ABC transporters have implications for cancer drug resistance and can also alter the toxicity of many drugs [17].

Finally, the epithelial cell adhesion molecule (EpCAM) is a structural MP that plays various roles in physiological processes and diseases such as cancer [25,26] and is known to be overexpressed in cancerous cells [27,28]. In cancers, including pancreatic, breast, colorectal, and prostate, the presence of circulating tumor cells (CTCs) in the peripheral blood was described [29]. These CTCs detach from primary tumors and enter the circulatory system, ultimately causing malignancies in distant secondary organs.

A critical factor that unites all the above-mentioned MPs is the outcome of their abnormal manifestation (i.e., mutated, overexpressed, etc.), upon which they can potentially serve as disease biomarkers detectable by diagnostic means [4]. In general, biomarkers must fulfill the following defining guidelines: (**i**) are relevant to the phenotype under investigation, (**ii**) can be assayed reliably, (**iii**) readily available (stable) for detection, and (**iv**) recognizable by current clinical methods [4]. Infectious diseases caused by pathogenic agents (e.g., viruses, bacteria, parasites, and fungi) can pose serious public health issues, so they get significant attention for clinical diagnosis using similar biomarker-based approaches [30,31,32]. For example, hemagglutinin (HA) is a well-known surface glycoprotein of the influenza virus [33] that attracted the development of targeted diagnostic procedures for HA-driven infections [34]. More recently, angiotensin-converting enzyme II (ACE2) and the SARS-CoV-2 spike and nucleocapsid proteins have all become important diagnostic and therapeutic targets in the fight against COVID-19 [35,36].

## 3. Diagnostic Technologies for Cell Surface Biomarkers

Conventional analytical techniques such as flow cytometry, mass spectrometry, liquid chromatography, and nuclear magnetic resonance have contributed vastly to the determination of analyte concentrations (including MPs) and their biochemical and structural characterization [37,38,39]. However, when diagnostic applications are considered, such techniques have major disadvantages, including limited availability, high costs, time consumption, and labor-intensive nature. They do not offer quantitative measurement of MPs in sensitive and reliable high-throughput assay formats. To tackle these challenges, antibodies raised against cell surface biomarkers are extensively used for cellular phenotyping, functional studies, and assessing expression profiles of MPs. In diagnostics, almost all hematopoietic diseases, for example, are detected using a panel of antibodies against a cluster of differentiation (CD) specificities, along with many other molecular imaging agents [5,40]. Antibodies are also used for the detection of CTCs based on the expression of epithelial markers, such as EpCAM [41,42]. However, this antibody-based approach can be limited as CTCs can lose their epithelial characteristics (e.g., EpCAM downregulation) due to epithelial-mesenchymal transition and can have antigen expression profiles that are shared with normal cells [29]. In this case, neither can antibodies detect CTCs nor discriminate between malignant and benign cells.

Flow cytometry is one of the most widely used and accessible antibody-based diagnostic methods to evaluate cell surface biomarkers [43]; other antibody-based technologies include Western blotting, enzyme-linked immunosorbent assays (ELISA), and tissue immunostaining [44]. Although immunoassays can be highly efficient and sensitive in detecting cell surface biomarkers, they have some limitations, including high cost, risks of interferences, the need for quality controls, labor-intensiveness, the requirement for special expertise and instrumentation, difficulties in quantifying MPs, and the inaccessibility to MPs as possible targets.

Biological sensors, on the other hand, have become ubiquitous platforms in many areas of applied sciences, including clinical diagnostics. These attractive and rapidly evolving tools are advantageous over many of the other traditional and bulk methods in several aspects, like providing results fast, on-site, and with minimal sample collection. In addition to their applicability in biomarker-based diagnosis, biosensors are applied to drug discovery, forensics, food control, environmental monitoring, and biomedical research [45,46,47]. In principle, ideal biosensors must be easy to use, portable, cheap, and up scalable for mass production. On the technical side, they must (**i**) respond to their targets (e.g., MP biomarker) in a highly specific and selective manner, (**ii**) be stable by resisting degradation, and (**iii**) output constant signals that are unaffected by ambient disturbances (e.g., temperature) to produce precise and reproducible readouts.

The high affinity of the recognition element to its target in the biosensor is a fundamentally important factor that contributes to the sensor’s reliable use, as it should interact with its target strongly. That is, high-performing biosensors are also defined by their ability to generate reproducible results with high sensitivity (i.e., low limit of detection) over a wide concentration range and in a linear fashion. Figure 2 summarizes the two main components of a typical biosensor and its mechanism. The recognition element interacts with the desired target based on its affinity, and then the interaction is converted into physically detectable signals by the signaling component [48,49,50].

## 4. Aptamers and Their Value

Aptamers, often referred to as ‘chemical antibodies’, are typically small-sized RNA or DNA nucleic acids made up of 20–100 nucleotides [51,52,53,54]. These nucleic acid aptamers are capable of binding to various desired targets such as small molecules (e.g., environmental and food contaminants, such as ochratoxin A and bisphenol A), proteins, viruses, as well as whole cells, in a specific manner and with high affinity. Affinities are often reported as the dissociation constant (Kd) that generally ranges from picomolar (pM) to micromolar (µM) values.

Aptamers have the propensity to fold into secondary structures like stems, loops, bulges, pseudoknots, G-quadruplexes, and kissing hairpins that help form unique three-dimensional (3D) shapes. The high affinity and specificity of aptamers against their cognate targets are attributed to their 3D structures that often undergo conformational changes upon target binding. The types of interactions that characterize aptamer-target binding include hydrophobic and electrostatic interactions, hydrogen bonds, van der Waals forces, shape complementarity, and stacking.

In 1990, three different groups contributed independently to the isolation of novel RNA motifs by an in vitro selection process [55,56,57]. Tuerk and Gold, particularly, achieved the first selection using a random pool of RNA sequences against T4 DNA polymerase [55]. The process of aptamer selection was termed the *Systematic Evolution of Ligands by Exponential Enrichment* (SELEX), an iterative procedure involving a certain number of selection rounds used to continuously enrich a random pool with the sequences potentially having the highest affinity against a target of interest. Briefly, a SELEX round constitutes three parts: (**i**) incubation of the target with the pool, (**ii**) separation of the bound from unbound oligos, and (**iii**) amplification of bound species for use in the next selection round. The synthetic oligonucleotide libraries used in modern selection studies typically contain about 10^15^ different random sequences and can be used for selection against virtually any target.

Compared to antibodies, aptamers are highly malleable [58,59]. That is, they are responsive to different intrinsic (by chemical modifications) or environmental (e.g., ions, temperature) cues, allowing them to undertake different forms, often without the irreversible loss of functional capacity. However, this also dictates a certain level of control to optimally utilize their capacities for given purposes. Nonetheless, RNA and DNA aptamers are prone to nuclease degradation; hence, stabilizing modifications might be necessary [60]. For example, RNAs have a half-life of minutes in human serum, while DNAs have a longer half-life of approximately 60 min. Post-SELEX optimization strategies such as chemical modifications, truncations, and mutagenesis provide, among other benefits (e.g., enhanced affinity), stability for the selected aptamers against degradation, favoring their storage and utilization in both diagnostic and therapeutic applications [61,62]. Although antibodies have been the most used probes in diagnostics to bind a broad range of targets with high affinities [63,64,65], studies have constantly shown that aptamers are worthy candidates to provide new opportunities and overcome issues related to the use of antibodies as diagnostic and therapeutic tools [51,59]. Table 1 lists different examples of aptamers selected for applications in diagnostics. The availability of such data presents new and possibly more desirable means outside the conventional to probe targets of interest. Indeed, like antibodies, aptamers recognize and bind their targets with high affinity and specificity. However, unlike antibodies, aptamers have little immunogenicity. Their toxicity is also low, and they can be selected against a broader range of targets, including non-immunogenic and toxic targets. In terms of size, aptamers are small molecules, allowing penetration through the blood-brain barrier and versatility to bind small epitopes inaccessible to the relatively larger antibodies. Regarding the production of antibodies, the process is generally done in vivo and involves the suffering of animals, yields batch-to-batch variations, and requires expensive and time-consuming downstream processes [31,66,67]. On the other hand, the selection of aptamers is a quicker, easier, and cost-effective process, and their chemical synthesis is highly reproducible and yields aptamers with high purity [59,67,68].

Antibodies are prone to denaturation (especially at high temperatures), frequently lose their functionality after use once or a few times, and are difficult to label at specific sites. Furthermore, assays that involve the utilization of antibodies often require immobilizations and extensive washing and are difficult to perform with homogeneity. Aptamers, on the other hand, are highly thermostable and renature easily after repetitive denaturation cycles, can be stored for longer periods of time (years), and can be repeatedly used without any loss of binding capacity. Their small size allows for achieving higher densities during immobilization. Aptamers can also be more sensitive compared to antibodies and, thus, are capable of differentiating target isoforms [69]. Finally, aptamers can be easily integrated onto solid supports such as polymer, carbon nanotube, and metals (e.g., gold) by means of electrostatic, hydrophobic, and covalent interactions as well as a self-assembled monolayer. Some linkers commonly used to carry out specific interactions for surface immobilization include biotin, streptavidin, avidin, neutravidin, and amine-, carboxyl- or thiol-groups [70,71,72].

SELEX is a continuously optimized process to obtain aptamers more efficiently and with minimal labor. Aptamers as analytical tools with the potential for target detection by their use in solution-based analyses, their integration into solid supports, or through in vivo applications, is a topic reviewed in greater detail in previous publications [68,73,74,75]. So, in the next sections of this review, we re-address this potential briefly, but in the context of MPs and the diagnosis of MP-related pathologies, with a focus on exploiting different physicochemical properties for the purpose of biosensing.

**Table 1 molecules-28-03728-t001:** Aptamers were selected recently against MPs (from 2011 to 2022). These MP-targeting aptamers can be integrated into aptasensor arrays for future diagnostic applications.

Aptamer Name	Target Protein	Backbone	Aptamer Applications	Reference
**EpCAM aptamer**	The transmembrane Glycoprotein epithelial cellular adhesion molecule (EpCAM).	DNA	Important candidate for deep-tumor treatment and drug delivery.	[76]
**Aptamer 1-717**	The transmembrane p24 trafficking protein 6 (TMED6)	RNA	Easily conjugatable aptamers with imaging reagents for β cell mass quantification and RNA therapeutics for the efficient non-viral transfection of human β cells	[77]
**C7 aptamer**	SARS-CoV-2 Spike (S) protein	DNA	Sensitive sandwich-FLAA test for SARS-CoV-2 detection	[78]
**CoV2-RBD-1C**	RBD protein S SARS-CoV-2	DNA	COVID-19 disease biomarker detection	[79]
**CA125 aptamer**	Blood tumor marker, carbohydrate antigen 125 (CA125)	RNA	Ovarian cancer detection	[80]
**I17 aptamer**	Intercellular Adhesion Molecule-1 (ICAM-1)	DNA	Early detection of atherosclerosis	[81]
**Np-A48 aptamer**	SARS-CoV-2 nucleocapsid protein (Np)	FQ-ssDNA	Diagnostic tools for COVID-19	[82]
**MSA_1_ and MSA_5_**	SARS-CoV-2 spike protein (S1 protein)	DNA	Diagnostic tools for COVID-19	[83]
**V11 and V21**	Enterovirus 71 (EV-A71 protein)	DNA	Early detection and treatment of EV-A71	[84]
**S6-1b**	Glioma SHG44 cells	ssDNA	Effective molecular diagnostic tools to detect early stages of malignant gliomas	[85]
**40L and A40s**	Ephrin type-A receptor 2 (EphA2)	RNA	Radiotherapy and chemotherapy for glioblastoma (GBM) treatment	[86]
**Aptamer-protamine-siRNA nanoparticle (APR)**	ErbB3 positive MCF-7 cells	RNAi	Genetic treatment for breast cancers	[87]
**V8 and V13**	Vibrio vulnificus	ssDNA	Detection of *Vibrio vulnificus*	[88]
**R3, R5 and R11**	Rice black-streaked dwarf virus (RBSDV) P10 protein	DNA	Potential for use in detection of RBSDV P10 protein in vitro and in vivo	[89]
**HBA1 and HBA2**	Avian influenza (AI) surface protein hemagglutinin	ssDNA	Effective molecular probes for diagnosing H5N1. Therapeutic inhibition of viral surface proteins	[90]
**SYL3C and NC3S**	EpCAM and N-cadherin as CTCs acquire mesenchymal marker	ssDNA	A promising tool for capturing CTCs from clinical samples	[91]
**M17**	MMP14	DNA	Tumor imaging, cancer therapy	[92]
**ApC1**	Colorectal carcinoma Caco-2 cells	DNA	Targeted therapy for colorectal cancer	[93]
**XQ-2d**	Membrane-bound CD71 protein of pancreatic cells	ssDNA	Promising tools for cancer biomarkers diagnosis and therapy	[94]
**Apt5**	PD-L1	DNA	Cancer cell imaging, CTC enrichment	[95]
**ECD_Apt1**	His-tagged human epidermal growth factor receptor 2 (HER2)–extracellular domain (*E. coli* system)	DNA	An effective, low-cost alternative to conventional anti-HER2 antibodies in solid-phase immunoassays for cancer diagnosis and related applications	[96]
**Heraptamer1 and Heraptamer2**	HER2 overexpressed in SKOV3 ovarian cancer cells	DNA	PET imaging of radiolabeled HER2 in vivo	[97]
**GL56**	Insulin Receptor (IR)	2′F-RNA	Inhibition of IR signaling, reduction of cell viability, and targeted therapies	[98]
**MRP1-CD28 bivalent aptamer**	Multidrug resistant-associated protein-1 (MRP1)	2′F-RNA	Reduction of cell growth in vitro and improved survival in vivo	[99]
**Apt02, Apt09, Apt10**	Integrin αv	RNA	A new SELEX method was developed: “Isogenic cell-SELEX”	[100]
**Integrin α6β4-specific DNA aptamer (IDA)**	Integrin α6β4	DNA	Imaging (confocal) applications and drug delivery	[101]
**MS03**	CD44/CD24	DNA	A promising molecular probe for breast cancer diagnostic and therapeutic applications	[102]
**HY6**	Extracellular domain of 20-amino acid HER2 peptide	Thio-DNA	Targeted therapy	[103]
**CLN64**	c-MET	2′F-RNA	Inhibition of tumor cell migration	[104]
**Sgc-3b and Sgc-4e**	Selectin L and integrin α4	DNA	Therapeutic intervention	[105]
**ACE4 aptamer**	MCF-7 cells	2′F-Py RNA	Internalization into cells upon binding to Annexin A2. Tumor targeting and imaging in vivo	[106]
**SDA**	E-and P-Selectin	DNA	Therapeutics for inhibition of cancer cell adhesion and metastasis	[107]
**Tutu-22**	EGFR	DNA	Novel targeted cancer detection, imaging, and therapy	[108]
**U2**	EGFRvIII	DNA	Radiolabeled imaging and diagnosis of glioblastoma	[109]
**Gint4.T**	PDGFR β	2′F-RNA	Inhibition of receptor signaling, cell migration and proliferation, and tumor growth in vivo. Induction of differentiation	[110]
**EP166**	Epithelial cell adhesion molecule-EpCAM (CD326)	DNA	Stem cell biomarkers	[111]
**SYL3C**	EpCAM	DNA	Novel targeted cancer therapy, cancer cell imaging, and CTC enrichment	[112]
**9C7, 11F11**	T-cell receptor OX40 T-cell	2′F-RNA	Increasing proliferation of T lymphocytes and production of IFN-γ. Potential for antigen-specific T cell stimulation together with dendritic cell-based vaccines (adoptive cellular therapy)	[113]
**CD28Apt2 and CD28Apt7**	Murine recombinant CD28-Fc fusion protein	2′F-RNA	Reduction of tumor progression and increased overall survival (in vivo). Enhancing vaccine-induced immune responses	[114]
**R-1, R-2, and R-4**	Human recombinant BAFF-R protein	2′F-RNA	Delivery of siRNA and combinatorial therapeutics	[115]
**Aptamer 32**	EGFRvIII	DNA	Delivery of chemical drugs and diagnosis	[116]
**Apt1**	GST-tagged human recombinant full-length CD44 protein	2′F-RNA	Therapeutic and diagnostic targeted delivery against stem cells	[117]
**Aptamer 2-2(t)**	ErbB-2/ HER2 in N87 cells	DNA	Acceleration of ErbB-2 degradation in lysosomes. Endocytosis-mediated inhibition of tumor growth in vitro and in vivo	[118]
**CD133-A, CD-133-A58, CD133-A35, CD133A21, CD-133-A15, CD133-B19**	CD133	2′F-RNA	Targeting cancer stem cells, molecular imaging	[119]
**C2NP**	CD30	DNA	Lymphoma Immunotherapy by activation of target oligomerization, downstream signaling, and apoptosis	[120]
**SQ-2**	Alkaline phosphatase placental-like 2-ALPPL-2	2′F-RNA	Targets both membrane-bound and secreted forms of ALPPL-2. Applications in diagnosis, imaging, and therapy	[121]
**CSC13**	CD44	DNA	Cancer detection, imaging, and drug delivery	[122]
**YJ-1**	Carcinoembryonic antigen	2′F-RNA	Inhibition of cell migration/invasion in vivo. Promotion of cell anoikis	[123]
**αV-1 and β3-1**	Integrin αvβ3	2′F-RNA	Multivalent aptamer isolation SELEX (MAI-SELEX) was applied	[124]
**HB5**	HER-2 peptide from the juxtamembrane region of HER2 extracellular domain	DNA	Drug delivery (Doxorubicin)	[125]
**cL42**	CD124 (IL-4Rα) recombinant ILR4α protein enzymatically cleaved	2′F-RNA	Reduction of tumor progression in vivo	[126]
**E1, B1, and C1**	N202.1A mammary carcinoma clonal cell lines expressing high levels of surface HER-2/neu	2′F-RNA	Drug delivery (Bcl-2 siRNA). Chemo-sensibilization and reduction of drug resistance	[127]
**C4-3**	Neurotrophin receptor, TrkB	2′F-RNA	Neuroprotective effects. Therapy of neurodegenerative disease	[128]
**GL21.T**	Axl	2′F-RNA	Interferes with cell migration and invasion, inhibition of spheroid formation and cell transformation, inhibition of tumor growth	[129]
**C2 aptamer**	CD71	2′F-RNA	Delivery of aptamer-functionalized siRNA-laden liposomes	[130]
**EpDT3-DY647**	Epithelial cell adhesion molecule-EpCAM (CD326)	2′F-RNA	Target stem cell marker for cancer nanomedicine and molecular imaging	[131]
**SE15-8**	ErbB2	2′F-RNA	High specificity to ErbB2 and not other members of the ErbB family. Applications in drug delivery and imaging for in vivo diagnosis	[132]
**-**	HER-2 overexpressing breast cancer cell line, SK-BR3	DNA	More effective probes against HER2-positive cells for diagnostic and therapy	[133]
**bsA17, bsA22**	Fcγ receptor III (CD16α)	DNA	A tumor-effective function of two aptamers linked into a bi-specific aptamer for cellular cytotoxicity	[134]
**CL4**	EGFR	2′F-RNA	Induces EGFR-mediated signal pathways causing selective cell death. Combined cetuximab-aptamer treatment induces tumor apoptosis in vitro and in vivo.	[135]

## 5. Aptamer-Based Biosensors for Diagnostic Applications to MPs

Many aptamers have been selected against MPs characterized as biomarkers (e.g., for cancer and stem cells) and bacterial or viral virulence factors [31,59,69]. Table 1 lists the MP-targeting aptamers suitable for use in diagnostics. Over three decades, methods for aptamer selection against MPs have ranged from the classical protein-SELEX, where the MP (or its soluble hydrophilic domain) is purified and used as a SELEX target, to the revolutionary cell-SELEX (whole living cells used as a target) [136] and its other more recent variants like TECS-SELEX (target is recombinantly overexpressed on the cell’s surface) [137], FACS-SELEX (selection using a fluorescence-activated cell sorter) [138], 3D cell-SELEX (3D cell cultures achieved by methods like magnetic levitation) [139], and in vivo-SELEX (selection in living organisms) [140]. A plethora of other advanced variants of SELEX can be successfully used to target MPs for more specific purposes. Although antibodies dominate the global diagnostic market, a growing number of aptamer-based biosensors (also known as aptasensors) are appearing at a rapid pace [31,51,141]. This is largely due to the aforementioned advantages of aptamers, which facilitate gaining more recognition and occasional preference over antibodies as an attractive class of small synthetic molecules for use in biosensors. Exceptionally, unlike antibody-based assays such as ELISA (enzyme-linked immunosorbent assay), aptasensors can also be readily multiplexed to achieve simultaneous measurement of several biomarkers for a more confident diagnostic evaluation. Nevertheless, in principle, each aptasensor is envisioned and designed specifically for the detection of a single and specific target of interest. Figure 3 illustrates the main successive steps of a functional aptasensor.

In 1998, Potyrailo et al. generated the first biosensor utilizing immobilized aptamers for the detection of free non-labeled thrombin in the solution [142]. The aptasensor detected thrombin at a concentration as low as 0.7 amol in a 140-pL interrogated volume. Today, based on defined physicochemical properties, aptasensors are commonly classified into electrochemical, optical, and mass-sensitive aptasensors (Figure 4). Particularly, aptasensor platforms that employ solid supports during sensing provide an opportunity for the measurement of analytes in real time [75,143,144,145]. In addition, current aptasensor developments pursue simultaneous measurement of multiple aptamer targets (multiplexing) as well as miniaturization.

### 5.1. Electrochemical Aptasensors

Electrochemical aptasensors are systems containing redox-labeled or label-free aptamers [146,147]. Redox labels can be either covalently linked to terminal groups of aptamers (e.g., enzymes such as horseradish peroxidase, metal nanoparticles such as gold or platinum, and other redox compounds such as ferrocene (Fc), methylene blue (MB), or anthraquinone) or non-covalently linked (e.g., intercalating MB and electrostatically interacting charged ions such as [Ru(NH_3_)_6_]^3+^). Although label-free aptasensor systems reduce the burden of additional aptamer labeling procedures, they may include other labeled molecules or redox ions as reagents in the solution (Figure 5).

Aptasensors exploit one of two phenomena (or their combination) to detect signals from aptamer-target binding: (**i**) the flexibility of aptamer structures where target binding induces precise conformational changes to its structure, and (**ii**) a binding-induced complementary strand displacement from a duplex structure. In electrochemical aptasensors, these properties allow reporting aptamer-target complexes as redox-tagged or untagged aptamers immobilized on conductive support to govern electrical communications with that support (an electrode) by means of the aptamer itself (tagged) or by other companion molecules (label-free). Figure 5 summarizes these concepts using redox-tagged surface-immobilized aptamers as an example. Electrochemical aptasensors typically work based on a “signal-on” or a “signal-off” setting. Signal-on refers to a positive readout signal where an electron transfer occurs between a redox label and the electrode. In contrast, signal-off refers to a negative readout signal where the electron transfer becomes hindered. The “on/off” shifts occur after target binding, and the outcome signal (on or off) is dictated by the induced aptamer conformational changes or strand displacement (or both). However, signal-off formats are difficult to work with (especially for diagnostics) [31] due to a decrease in amperometric response after the target interaction with its aptamer. To overcome this problem, signal-on formats were designed wherein electrical communication takes place only after aptamer-target interaction [148]. Using these electrochemical sensing systems, target concentrations can be correlated with measurements of electrochemical features after target binding.

Electrochemical transducers for the detection of analytes are based on techniques such as amperometry, potentiometry, impedimetry, and field-effect transistors (FETs). Sassolas et al. reported that amperometric and impedimetric modes of transduction had been the most used in aptasensor development [147]. Electrochemical aptasensors have successfully detected proteins, such as thrombin, platelet-derived growth factor (PDGF), and immunoglobulin E, as well as small molecules, such as cocaine, adenosine triphosphate (ATP), theophylline, aminoglycosides, and potassium ions (K^+^). The first electrochemical aptasensor was constructed in 2004 with a sandwich-based design for the detection of thrombin, where a thiolated aptamer immobilized onto the gold electrode and an added glucose dehydrogenase-labeled aptamer were used [149].

For the detection of MPs, various notable examples using electrochemical aptasensors are reported. One example involves the measurement of Mucin 1 (MUC1) protein, a transmembrane protein expressed in epithelial cells. Its expression level is known to increase dramatically in most human epithelial cancers, and so it serves as an important biomarker for cancer detection [150]. Therefore, various techniques, including ELISA, as well as colorimetric, fluorescence, electrochemiluminescence, and electrochemical methods, have been developed for MUC1 detection. For example, an ultrasensitive electrochemical aptasensor was fabricated by a gold electrode layer and three different hairpin DNA aptamers (HP1, HP2, and HP3) [151]. Briefly, non-immobilized HP1 complexes with MUC1 lead to the opening of its hairpin structure. The exposed segment then attacks immobilized HP2 to form a double-stranded structure with a newly exposed segment. This segment could then also hybridize with HP3 modified with PtPd bimetallic nanoparticles (PtPdNPs). The complex comprising MUC1 and the HP1 aptamer is finally released by strand displacement. The bimetallic nanoparticles were employed as mimic peroxide probes catalyzing the oxidation of tetramethylbenzidine (TMB) by H_2_O_2_ and leading to the amplification of the electrochemical signal. The aptasensor had a linear response from 100 fg/mL to 1 ng/mL and a limit of detection (LOD) as low as 16 fg/mL MUC1 and demonstrated satisfactory results with serum samples.

Another example of MP detection by an electrochemical aptasensor involves the sensing of exosomes carrying the transmembrane protein CD63 [152]. CD63 is a tetraspanin protein considered to be a biomarker of exosomes—extracellular vesicular bodies that have the potential for use in cancer diagnosis as they are correlated with tumor antigens and antitumor immune responses. The aptamers were immobilized onto gold electrodes and incorporated into a microfluidic system to minimize sample volume. MB-probed strands were hybridized with the anchored aptamers, and after binding the aptamers to CD63 carried in exosomes, the DNA duplexes released the antisense strand carrying redox reporters, ultimately leading to a decrease in the electrochemical signal. The sensor showed a detection limit of 1 × 10^6^ particles/mL and a linear range extending towards 1 × 10^8^ particles/mL, demonstrating its higher sensitivity compared to conventional exosome quantification methods.

For acute lymphoblastic leukemia, an oncological disease prevalent in children, a redox-labeled electrochemical aptasensor that specifically detects the cancer biomarker protein tyrosine kinase 7 (PTK7) was developed [153]. Detection was performed using Jurkat leukemia cells that overexpress PTK7, and for this purpose, two aptasensors with different redox markers (MB or Fc carboxylic acid) were described. Their sensitivities were compared using the target Jurkat cells at a concentration range of 50–5000 cells/mL and by differential pulse voltammetry. Similar LOD values were obtained for both aptasensors with 37 ± 6 cells/mL for Fc-labeled aptamers and 38 ± 8 cells/mL for MB-labelled ones.

### 5.2. Optical Aptasensors

Optical bioassays have widely benefitted from the use of aptamers as biorecognition elements. Optical analyses can readily achieve multiplexing for the simultaneous detection of several analytes [154]. Many powerful and advantageous optical techniques were utilized for aptasensing (e.g., luminescence, surface plasmon resonance, surface-enhanced Raman scattering, optical fibers, etc.). For example, both luminescence resonance energy transfer [155,156,157,158] and electrochemiluminescence [159,160] were used as the basis for the development of aptasensors to detect exosomes, cancer biomarkers, and bacterial pathogens. Towards surface plasmon resonance, despite the common categorization of the method as an optical sensing technique, in this review, we present the method from the direction of mass-sensitive detection (see Section 5.3). Surface-enhanced Raman spectroscopy (SERS) for analytics has seen great interest in different areas of science, as witnessed by the thousands of publications over the decades [161,162]. The rapid and ultrasensitive technique has recently gained considerable momentum with regard to the detection of respiratory viruses for the purpose of point-of-care testing, especially after the COVID-19 pandemic. When combined with the specificity and versatility (here, Raman-active labeling) of nucleic acid aptamers, the technique was able to reach low limits of detection (that could challenge the limits of PCR techniques) as shown for SARS-CoV-2 [163,164] as well as the influenza A virus [165,166]. SERS was also applied recently for the sensitive detection of cancer biomarkers, such as carcinoembryonic antigen (CEA) [167]. In particular, Zhang et al. have shown valuable improvement in the utilization of SERS through the detection of cancer cell-derived exosomes in clinical serum samples [168]. Their ratiometric SERS aptasensor (for HER2 and EpCAM), as opposed to conventional SERS, demonstrated high accuracy for target identification (considering the heterogeneity of cancer exosomes) and ultrahigh sensitivity for the early detection of cancer in patients and without any need for nucleic acid amplification.

Biosensors integrating optical fiber technologies have also been well-described in research articles and numerous reviews and have acquired special interest recently [169,170]. These fiber optic biosensors are shown to have numerous advantages that include but are not limited to rapid, ultrasensitive, label-free, real-time, low-volume, and on-site detection, as well as miniaturization. Janik et al. have recently described optical fiber aptasensors based on a microcavity in-line Mach-Zehnder interferometer (μIMZI) for the detection of pathogenic bacteria (*E. coli* O157:H7) [171] and, for the first time, virus-like particles (modeled by SARS-CoV-2), achieving detection limits of 10 cfu/mL and 1 ng/mL, respectively [172]. Other optical techniques based on resonance scattering, dynamic light scattering, and ellipsometry were also integrated into the development of aptasensors [173]. However, despite impressive and promising advancements in optical sensing technologies with aptamers as biorecognition elements, the indispensable fluorescence and colorimetric techniques are the most popular and employed methods [148,174]. Notably, water-soluble conjugated polymers have specifically accumulated special interest due to valued properties (optical, electronic, solubility, brightness, (photo)stability, low cytotoxicity, etc.) that effectively mediate interdisciplinary biological applications [175]. Optical reporting with aptamer-conjugated polymers is simple, cost-effective, can be fluorescence- or colorimetry-based, and is achievable in both a labeled and an unlabeled manner [176,177].

#### 5.2.1. Fluorescence-Based

Fluorescent detection is widely used as aptamers are easily labeled with fluorescent dyes and have many available fluorophores and quenchers (Figure 6). Fluorescence also has the capability of real-time detection. The commonly used molecular beacon systems also employ aptamers (aptabeacon) and therefore consist of the aptamer in a hairpin structure and end-labeled with a fluorophore and a quencher. During target absence, the stem-loop is in a closed position, and the quencher is in close proximity to the fluorescent dye. The binding of the target disrupts the stem and moves the fluorophore away from the quencher, yielding a fluorescence signal (Figure 6). In another format of fluorescence-quenching-based probing, a fluorophore-labeled aptamer is present in a duplex structure with a quencher-labeled complementary DNA sequence.

More complex structures involving quaternary structural rearrangements where aptamers assemble and disassemble have also been developed to achieve fluorescence signaling [178,179]. Additionally, nanomaterials, such as quantum dots, gold nanoparticles (AuNPs), graphene oxide, polymer nanobelts, and coordination polymers, are more recent candidates used over traditional materials to attain improved fluorescence-quenching effects. As an alternative to fluorescence quenching approaches, Förster resonance energy transfer (FRET) is an approach that is based on an exchange of energy between two fluorophores—a donor and an acceptor. Although dual-labeled oligonucleotides are continuously in development, FRET optimization for quantification purposes is difficult to achieve due to the numerous factors affecting FRET. Moreover, fluorescent detection, whether using fluorophore-quencher pairs or FRET pairs, is difficult to apply against targets in their native environments within complex biological samples due to background signal interferences. A possible solution to circumvent this problem is to use fluorophores that shift their fluorescence wavelength emission to bypass signal interferences.

An example of the detection of MPs by fluorescence-based optical aptasensors is provided by Bahmani et al. achieving the detection of CD44 exon v10 [180]. CD44 is an integral transmembrane protein expressed in breast cancer cells in various isoforms. The important exon v10—one of the isoforms of the 20-exon CD44 single gene—plays a critical role in promoting the progression and metastasis of breast cancer. It was discovered that the isoform mediates the formation of a molecular complex on breast cancer cell surface with the RTK, EphA2 (receptor for Ephrin1), a key player in the development and metastasis of many malignant tissues, thereby demonstrating the need for effective targeting of v10 isoform for detection and prognosis. In their study [180], v10-specific DNA aptamers were used to synthesize aptamer-templated fluorescent metal nanoclusters (Apt-NCs). NCs of silver, gold, and copper were prepared using different aptamer templates, and the synthesized Apt-NCs were confirmed for accuracy and quality by UV-Vis, transmission electron microscopy, and fluorescence spectrometry. It was shown that compared to native aptamers, modified ones had formed more stable and brighter NCs that are sufficient for cell detection assays using different cultured cell lines. In comparison to other NCs, aptamer-modified copper NCs (M-Apt4-CuNCs) have shown, by their fluorescent response, a higher efficiency in tracing CD44 v10 on the cell surface and a proper correlation with concentrations of the target on the cells. The specific and sensitive aptasensor has a detection limit of 40 ± 5 cells/mL.

Recently, a FRET-based detection system was constructed by utilizing novel fluorescent probes and graphene oxide to detect H5N1 influenza A virus HA, found as trimeric spikes on the viral membrane [181]. The highly conserved regions of HA are crucial for viral function and replication and represent an important biomarker for diagnosing H5N1 infections. Synthesis of sub-20 nm sandwich-structured upconversion nanoparticles (SWUCNPs or UCNPs) improved energy transfer efficiency and allowed control of the emitter in a thin shell [181]. The lanthanide-doped UCNPs are advantageous over conventional downconversion fluorescent probes. They can be excited with near-infrared light, avoiding interference from background biological fluorescence, enhancing stokes shift, lifetime, and light stability, and narrowing the emission spectrum. This translates into an improved signal-to-noise ratio in biological detections that use fluorescent labels. FRET was achieved through the π–π stacking interaction between the aptamer and graphene oxide, bringing the fluorescent probes closer and realizing FRET-induced fluorescent quenching. This interaction is eliminated by the formation of the H5N1 HA-aptamer complex, resulting in the detection of a fluorescence signal. The novel aptasensor shows a linear response from 0.1 to 15 ng/mL of HA concentrations, and its LOD was 60.9 pg/mL. Furthermore, the method was applied with human serum and had a linear range of 0.2–12 ng/mL with a LOD of 114.7 pg/mL.

#### 5.2.2. Colorimetry-Based

Optical sensors that yield an observed colorimetric signal (color change) in the presence of a target analyte utilize different approaches for signal detection, such as ligand-receptor interactions (fluorophores and chromophores), fluorescence quenching (organic dyes, polymers, etc.), enzyme and mimetic nanoenzymes, and more recently, gold nanoparticles (AuNPs) [182]. In addition to sensitivity and selectivity, colorimetric detection offers the observation of color changes by the naked eye, a minimal or complete lack of instrumentation, and better management of on-site and real-time detection [183]. Detection with colorimetric assays can also be performed easily in solution because they do not require any platform immobilizations or sophisticated equipment [184]. Labeled or unlabeled AuNPs have been used extensively in colorimetric assay applications [185,186,187]. Specifically, for colorimetric aptasensors, AuNPs are one of the most used color reporting factors. They are attractive due to various unique properties such as high biocompatibility, chemical stability, strong localized surface plasmon resonance absorption, and high extinction coefficient in the visible region. Colorimetric assessments using AuNPs can be carried out in one of two ways: an analyte-induced assembly (aggregation) or an analyte-induced disassembly (disaggregation) of AuNPs [188,189]. When dispersed, AuNPs are red in color but turn purple or blue when aggregated (Figure 7). Additionally, aggregation dynamics of AuNPs in non-crosslinking assays are affected by factors such as electrolyte concentration, making the assay often simpler, faster, and more desirable than the cross-linking approaches. Colorimetry combining the use of DNA aptamers and metal nanoparticles (especially gold) has been of substantial interest since 2012 [182,190,191]. These DNA-functionalized AuNPs are used for the development of many aptasensing systems as the nucleic acid aptamer molecules are properly and functionally immobilized on the AuNP surface by the various immobilization methods available. Proper immobilization includes the optimal surface density of the immobilized aptamers that allows conformational flexibility of the aptamer for target binding events.

A simple and rapid colorimetric detection assay was developed for Salmonella typhimurium, a food-borne pathogen that causes intestinal infections [192]. In this study, chemically inert, easily separable, and catalytically stable Fe_3_O_4_ magnetic nanoparticles (MNPs) were used for their peroxidase-like activity to promote, in the presence of H_2_O_2_, the oxidation of 3,3′,5,5′-tetramethylbenzidine (TMB), the colorimetric substrate. This colorimetric reaction mediated by the peroxidase-like activity of MNPs generates the color change visible by the naked eye in the solution. DNA aptamers were used to reduce the peroxidase activity of MNPs by adsorbing to their surface, thereby blocking it and leading to MNPs aggregation and a decrease in their colorimetric properties as well. In the presence of *S. typhimurium* in the solution, the DNA aptamers—having a high affinity to an outer membrane protein on the bacterial surface—re-expose the MNPs surfaces for the enzyme-like reaction to take place, leading to a color change that is detected spectrophotometrically.

In a related sensor design, a simple, sensitive, and selective colorimetric aptasensor platform was fabricated [193]. A peroxidase-mimicking hybrid material, ZnFe_2_O_4_/reduced graphene oxide (rGO), was synthesized and used for its peroxidase-like activity (TMB oxidation by H_2_O_2_) that is enhanced compared to the individual metal oxide nanomaterial and carbon-based nanomaterial. Biotin-modified aptamers were immobilized by avidin on a microplate to act as the capture probe for *S. typhimurium*. In the presence of the bacterium, a sandwich-type complex is formed where the ZnFe_2_O_4_/rGO hybrid nanocomposites conjugated to another aptamer act as the signal probe by binding to the bacterium captured by the immobilized aptamer. An optical signal is generated in the presence of TMB-H_2_O_2_ due to a typical blue color formation. The aptasensor exhibited a LOD of 11 cfu/mL and a linear range from 11 to 1.10 × 10^5^ cfu/mL.

A more recent example of colorimetric aptasensors is demonstrated for the quantitative profiling of surface proteins located on extracellular vesicles (EVs) [194]. EVs are membranous structures that can originate not only from the plasma membrane but also from the endomembrane systems of almost every type of cell. Wang et al. employed EVs derived from two breast cancer cell lines (MCF-7 and MDA-MB-231) as well as from the plasma of a breast cancer patient and a healthy volunteer. They presented a simple, efficient, and wash-free colorimetric aptasensor based on the controlled growth of aptamer-functionalized AuNPs for the detection of EVs. The two cell lines were used as suitable models due to the variable expression of CD63, EpCAM, and MUC1 in their EVs. The aptamers (AptaCD63, AptaEpCAM, and AptaMUC1) used to modify the AuNPs surface individually will bind with Au^3+^ after adding the Au growth reagent, and the color of the solution changes from light to deep red in the absence of the target. In the presence of the target (EV surface proteins), Au aptasensors bind to it, changing the color from red to blue by induction of electronic coupling between the grown nanoparticles. The color changes are detected by the naked eye or by UV-Vis spectrometry, and they depend on the surface protein expression levels on EVs. The developed colorimetric assay can reach a LOD as low as 0.7 ng/µL based on EpCAM expression on MCF-7 EVs. Compared to conventional ELISA (LOD = 77 ng/µL), the LOD of the colorimetric assay therein is significantly lower and is comparable to recently developed methods.

### 5.3. Mass-Sensitive Aptasensors

Mass-sensitive biosensors can be defined as devices that can measure a property that is related proportionally to mass associated with or bound to the sensor’s sensitive surface comprising capture probes [148]. Aptamer-based mass-sensitive biosensors are typically label-free bioassays [195]. This has numerous pros, including saving biosensor development resources (e.g., time and costs), conducting analyses in a shorter time, retaining the high affinity of aptamer sensing, and decreasing non-specific interactions. In fact, mass-sensitive sensors are seen as the most advanced among all categories of biosensors [196]. They can achieve the highest sensitivities and are rapidly gaining interest with the research progress in physical phenomena and the mechanics of biomolecules.

#### 5.3.1. Evanescent Wave-Based

The physical phenomenon, surface plasmon, is excitable by the evanescent wave, giving rise to the effect known as surface plasmon resonance (SPR) [197]. SPR is commonly used to analyze biomolecular interactions and measure the affinity of interactions and analyte concentrations. Although the SPR method is known for use as an optical sensing technology and evanescent wave-based aptasensors can be classified as optical aptasensors, SPR devices are capable of sensing mass changes through the accompanied changes in refractive index on their surface [198,199]. Evanescent wave-based aptasensors, therefore, can utilize SPR to sense aptamer-target interaction and binding. For multiplexing purposes, aptamer arrays are analyzable via SPR imaging, which also allows real-time detection and raises the efficiency of detection to high-throughput levels [200]. In SPR-based aptasensing, the sample is injected through the flow cell, and target binding to the aptamers immobilized on the plasmon surface creates mass accumulation on the surface. This results in changes on the surface for the refractive index and SPR angle upon excitation by the electromagnetic waves, giving information about the density and mass changes on the typically gold or silver layer [148,201].

In 2019, a sensitive and selective aptasensor based on SPR was developed for the direct and quantitative detection of cancerous exosomes with signal amplification via dual gold nanoparticles [202]. The concentration of the exosomes was determined based on the changes in the SPR resonance angle. Using the SPR-based sensor, a LOD value of 5 × $10^{3}$ exosomes/mL was achieved, which was a 10-fold improvement over commercial ELISA. Moreover, the SPR aptasensor differentiated between exosomes from MCF-7 breast cancer cells and MCF-10A normal breast cells and was able to detect exosomes in 30% fetal bovine serum. Other efforts in SPR aptasensor designs aimed toward the detection of circulating cancerous biomarkers, an all-fiber plasmonic aptasensor was developed for the challenging detection of CTCs known to exist in very low concentrations in the blood [203]. Particularly, breast cancer CTCs were targeted using aptamers selected against mammaglobin-A present on the surface of the CTCs. Label-free, real-time detection achieved a LOD of 49 cells/mL and 10 cells/mL using gold nanoparticles for signal amplification.

For pathogenic agents, a localized SPR (LSPR) biosensing platform was presented for highly sensitive detection (single bacterial cell detection) of whole cells of the microorganism *Pseudomonas aeruginosa* strain PAO1 [204]. The concentrations of the bacteria captured by the surface-confined aptamers were shown to be linearly correlated to the redshift in wavelength of the extinction maximum of LSPR that results from aptamer target recognition (range = 10–$10^{3}$ cfu/mL). The LSPR sensor was also described to be rapid (∼3 h) and selective for the target over other strains of *Pseudomonas* and *E. coli* and to have an excellent shelf life (up to 2 months) and a clinically relevant dynamic range.

Severe acute respiratory syndrome coronavirus 2 (SARS-CoV-2) emerged at the end of 2019 and caused a global pandemic. For rapid diagnosis, an SPR sensor was developed using an aptamer selected against the receptor-binding domain of the spike glycoprotein. The spike protein antigen was optically detected (without amplification systems) via a previously selected aptamer and with a LOD of ∼37 nM [205]. The aptamer was immobilized on a short polyethylene glycol interface on a D-shaped plastic optical fiber probe with a gold nanofilm deposit [206]. The specificity of the sensor was established by testing with non-sense aptamer sequence and non-specific proteins (bovine serum albumin, AH1N1, hemagglutinin, and spike protein from middle eastern respiratory syndrome (MERS) coronavirus). A preliminary test was also carried out using diluted human serum (1:50 *v*/*v*), and the result was observed by the resonance shift (LOD = 75 nM). Such data encourage the use of these rapid, sensitive, and low-cost sensing devices in diagnosing many diseases, including the ongoing pandemic caused by COVID-19.

The detection of whole cells (and supramolecular structures, in general) has certain challenges when using SPRs [207]. Nonselective binding leading to refractive index changes is one challenge that was surmounted by employing reference flow cells to offset the effect caused by the nonselective binding [208]. The sensing range is a second limitation of SPR use in cell detection. Whilst this range is typically 200 nm in SPR, the dimensions of a cell are in the micron range. In this case, using long-range SPR increases the depth over 1000 nm by changing the incident light wavelength, thus enhancing the sensitivity towards cell detection [207].

#### 5.3.2. Acoustic Wave-Based

Acoustics-based sensors are devices that commonly operate based on the piezoelectric effect of a given functionalized material (called a substrate) and have been researched extensively over the past years for their low cost, high sensitivity, and portability, and their applicability in biorelevant detection [209]. These platforms can effectively determine protein affinity and monitor protein-protein interactions and complex formation on the piezoelectric surface [201].

Biosensing research has largely focused on two types of acoustic wave-based sensors. First, bulk acoustic wave-based sensors known as Quartz Crystal Microbalances (QCM) use thickness-shear mode (TSM) vibration with the vibration of the whole substrate (a quartz crystal). The generated acoustic wave propagates through the complete bulk of the piezoelectric crystal that is patterned by electrodes on two sides of the crystal. Mass loading on the sensor’s surface alters the natural frequency of the propagating resonance wave, and the frequency shift is measured by the QCM [201]. The resonance frequency of the QCMs ranges from 10 to 50 MHz; however, higher frequencies are always desired because of the accompanied increase in mass sensitivity [210]. Unfortunately, QCM devices become too thin and fragile for practical use at higher frequencies. Nevertheless, early research has shifted focus to QCMs since (unlike the second type of acoustic sensors) they were initially seen as a better option for biosensing applications due to their ability to not suffer from large attenuation caused by the introduction of the aqueous biological sample.

The second type of acoustic wave-based sensors is surface acoustic wave (SAW) devices that also perform based on the piezoelectric effect. These highly sensitive devices produce and detect acoustic waves using interdigital transducers on the surface of a piezoelectric crystal [71,211]. It is for this reason that propagation of the acoustic wave on these devices takes place on the surface of the crystal rather than the bulk of the substrate. Due to the different modes of wave propagation and the operation at a higher frequency range (100 MHz to GHz), SAW devices tend to be more sensitive to any surface changes (e.g., mass loading, viscosity, and conductivity changes, etc.) compared to QCMs [209,210,211]. SAW-based sensors had to go through many series of development (including sensing-surface modifications) involving relief of the extreme attenuation experienced in the presence of aqueous samples (e.g., buffered samples) before adaptation to the highly sensitive detection of biological analytes. One common approach in SAW biosensing involves using Love wave-type SAW sensors in which the Love waves propagate in the form of horizontal surface acoustic waves, reducing energy dissipation and increasing surface sensitivity [212]. The aptamer-based Love wave sensor chip designed by Schlensog et al. employing immobilization on a gold layer on the quartz crystal allows label-free sensitive detection of human α-thrombin and HIV-1 Rev peptide with a LOD of approximately 75 pg/cm^2^.

For the detection of MPs by the QCM method, a group reported the use of a DNA aptasensor targeting HER2 receptors for the detection of HER2-positive breast cancer cells, one of the most aggressive and fatal cancer cells [213]. HER2 is an important receptor and onco-marker that belongs to the epidermal growth factor receptor (EGFR) family and is overexpressed by 15–30% in breast cancer. To mitigate the time consumption and high degree of false positives in diagnostic HER2 detection, a label-free acoustic QCM aptasensor (referred to as TSM in the paper) was fabricated for the detection of SK-BR-3 breast cancer cells. Biotinylated DNA aptamers were immobilized at the neutravidin layer chemisorbed at the gold surface of the TSM transducer. Sensitive and specific detection was achieved by the decrease and increase in resonant frequency and motional resistance, respectively. Achieved LOD values were 1574 cells/mL and 1418 cells/mL with two different aptamers. A higher sensitivity was attained using aptamer-modified 20 nm AuNPs conjugated to streptavidin. A LOD of 550 cells/mL was subsequently observed. A recent review elaborately depicts QCM biosensors and the various recent developments and opportunities [214].

For SAW-based biosensors, much of the recent work done to detect MPs is seen taking the immuno-sensing rather than the aptasensing approach. This work focused on the detection of the cancer biomarker carcinoembryonic antigen [215], influenza A virus [216], and exosomes [217]. For SAW-based aptasensing, Chang et al. delivered an article describing a novel sensing technique based on a 2 × 3 model of a leaky surface acoustic wave (LSAW) aptasensor, a SAW-sensor variation developed with the progress in microelectronics and acoustics [218]. Their novel LSAW aptasensor array was targeted for the label-free, specific, and high-sensitive detection of CTCs. The 2 × 3 model design was developed to improve the efficiency of detection. MCF-7 tumor cells overexpressing MUC1 on their surface were used as a model target; aptamer-cell complexing and subsequent mass loading led to a phase shift, and a LOD as low as 32 cells/mL was achieved for MCF-7 cells.

#### 5.3.3. Mechanical Cantilever-Based

Mechanical cantilevers are alternative signal transducers utilized in areas such as cantilever-based sensors [219,220]. Such sensors offer a highly sensitive, quantitative, real-time, and label-free target detection by a platform displacement-based mechanism. Cantilever sensors are applicable in a wide range of areas, including biological (e.g., cells, proteins, and other biomolecules) as well as chemical detection [221,222,223,224]. These sensors are relatively small-sized and, accordingly, have low noise, resulting in higher resolutions, and can attain high scalability, which supports their use in a multiplexed manner for point-of-care testing. In simple terms, a cantilever can be considered a miniature adaptation of a diving board [71], which is typically gold-coated when nucleic acid immobilization is undertaken [201]. Cantilever-based aptasensors create surface stress by adsorption of the analyte onto the aptamer-functionalized surface [32,225]. A difference in surface stress between the top and bottom layers is created as a result of mass loading on the sensor cantilever. A difference in surface stress between those two layers creates the upward or downward force that is referred to as “deflection” or “bending” acting on the cantilever board (Figure 8). Fritz elaborately illustrates the different cantilever sensor modes of operation and the different cantilever surface molecular interactions that dictate the directionality of deflection in the surface stress mode [226].

In general, cantilever biosensors operate under one of two modes, referred to as the static- and dynamic modes. In the static mode, the length of the surface stress-induced deflection changes as a result of the degree of aptamer-target biomolecular interactions taking place in the cantilever system, where this deflection is measured by the target-sensing platform. Moreover, a blocked (non-binding) companion cantilever that functions as the reference accounts for the deflection caused by nonspecific interactions and can be utilized to minimize the effect of nonspecific binding and other disturbances (e.g., thermal drift) and to accurately determine the differential deflection (Figure 8) [225]. On the other hand, in the dynamic mode, also known as the resonant mode, binding-induced changes in the cantilever resonant frequency take place by virtue of the mass change or, occasionally, the stiffness change in the sensor cantilever [227]. In other words, the measured resonant frequency in the dynamic mode is a function of the mass of the cantilever. Although higher resolutions are generally achieved by cantilever sensors employing the dynamic mode, cantilever dynamics are strongly sensitive to fluid effects (e.g., viscous damping), which can severely affect the sensor sensitivity [220,227]. In contrast, in the static mode, the sensing medium and the environment are not severely limiting to the reliability of the detected signals. The extremely small but detectable cantilever mass changes (whether by bending in the static or resonance change in the dynamic mode) are measurable by different means [220]. The optical beam deflection technique is the most common in the static mode. Variations in the optical technique include the detection of changes in optical diffraction and interferometry. Beyond optical techniques, others measure changes in piezoelectric, piezoresistive, capacitive, or contact area of resistance [219]. It is also noteworthy that cantilever mass changes are not only detectable by direct measurement of the changes in deflection length (static mode) or resonance frequency (dynamic mode) but, as illustrated by Ziegler, can also be measured through the changes in the force constant, or in calorimetric effects (e.g., temperature and specific heat capacity (ΔQ)) on a bending bimetallic cantilever [219].

Li et al. developed an aptamer-based microcantilever array biosensor for the simultaneous optical readout and real-time detection of two biomarker analytes in solution [228]. One of the analytes was the carcinoembryonic antigen (CEA), a cell surface glycoprotein overexpressed in virtually all colon cancer cells, and the other was α-Fetoprotein (AFP), a non-membrane-bound oncofetal plasma glycoprotein. The signal of the interaction of the aptamers’ self-assembly immobilized on the gold-surface cantilever and their targets was scaled up by employing an aptamer-antigen-antibody sandwich assay. The multiplexed assay achieved a detection limit of 1.3 ng/mL and 0.6 ng/mL for CEA and AFP, respectively, demonstrating great potential for the simultaneous detection of multiple targets in clinical diagnosis settings.

Another microcantilever-based aptasensor was developed for the detection of the transmembrane epithelial tumor biomarker MUC1 [229]. In this study, thiolated aptamers were used to functionalize the cantilever surface, which was exposed to a MUC1 solution for binding, and the deflection of the sensing cantilever was measured by a position-sensitive detector. For MUC1 concentrations in the range of 5–500 nM, linearity and a low detection limit of 0.9 nM were achieved. In this study, the overexpression of MUC1 on the surface of breast cancer cell line MCF-7 was also diagnosed using the same microcantilever aptasensor, and the MCF-7 cells displayed an interaction with the surface aptamers. However, MCF-7 cells concentration in the range of 2.0 × 10^3^ to 5.4 × 10^4^ cells/mL demonstrated a linear relationship with cantilever deflection; a detection limit of 213 cells/mL was obtained. Therefore, as a novel approach, the microcantilever aptasensor can be used for early-stage detection of cancer biomarkers and whole cells.

## 6. Summary and Future Scope

According to the Biomarkers Definitions Working Group convened by the National Institute of Health (NIH), a biomarker (e.g., DNA, RNA, proteins, small metabolites, etc.) is a measurable indicator for normal biological or pathogenic processes and responses to pharmacological intervention [230]. Therefore, biomarkers have important applications in modern healthcare systems where they are used in the diagnosis of diseases and their extent, as well as their prognosis [231,232]. Other related biomarker applications can be seen in routine health checkups and monitoring of patient health status. As summarized in this review, membrane-associated proteins represent essential biomarkers for disease detection and treatment as they primarily participate in the pathogenesis of inflammatory, cardiovascular, and neurodegenerative diseases. In many cancers, accumulation (high expression) and accessibility of MPs on the cell surface make the cell highly recognizable. For these reasons, a considerable number of MPs are targets of therapeutic drugs and diagnostic molecules in the biomedical field nowadays. Targeting these biomarker surface proteins facilitates devising appropriate diagnostics and therapeutic intervention plans to prevent disease progression and other serious complications that may potentially emerge in patients.

Aptamers are unique ‘recognition molecules’ due to their high affinity and specificity towards their targets. Many studies have been carried out on the generation of aptamers against MPs, and a far larger number of studies have investigated aptamer applications in diagnostic biosensing for MPs. Clearly, properties of aptamers, such as their structural flexibility, small size, reusability, and cost-effective and easy preparation, allow their use as detection tools in biosensors. The preclinical literature dealing with aptasensors is vast; the recent global COVID-19 pandemic has further boosted the number of papers in the field. Indeed, as early as the first aptamer selection attempts, the ‘aptasensing’ strategy was extensively researched and developed to demonstrate the potential versatility (i.e., fast, reliable, and sensitive) in the detection of clinical targets like small molecules, cells, and viruses.

Throughout this review, we cited work that shows how aptasensors (electrochemical, optical, or mass-sensitive) (Figure 4) can be more desirable (e.g., delivering lower detection limits and stability) than commonly used antibody-based detection methods like ELISA; however, aptasensors are yet to be ready for standardized global-scale use in clinical applications. Most studies are only upstream of the technology development pipeline (proof-of-concept at preliminary stages). Nevertheless, several aptamers have entered clinical trials and, in some instances, have passed and achieved commercialization. The global aptamer market is projected to grow annually at a compound rate of 28.2% (2018–2025) and from $723.6 million to $5.0 billion (2016–2025) [233,234]. For example, among all therapeutic aptamers that successfully entered some phase of the clinical trials, the RNA aptamer Macugen^®^ (Pegaptanib sodium) for the treatment of wet age-related macular degeneration is the sole aptamer-based drug approved by the Food and Drug Administration in 2004. On the diagnostic front, the number of clinical trials on aptamers (and aptasensors) is limited. Currently, about 50 aptamer-related national clinical trials are ongoing, while only 11 are aimed at diagnostic purposes [234]. However, of 40 global companies actively engaged in preclinical and clinical aptamer research, several have successfully commercialized their aptamer-based biomarker diagnosis products after undergoing rigorous clinical testing [235].

Aptamers represent potent and unique alternative device options to be established as mainstream synthetic MP probes, thereby revolutionizing aptamer-based point-of-care diagnostics and occupying niches in the market and the clinic that are hardly taken up by other molecular devices [233]. A major reason behind the commercial lag of these exciting molecules stems from the huge industrial and financial investments in and commitments to antibodies, breeding a global reluctance to shift to new paradigms. Furthermore, this reluctance is accompanied by a wide-scale lack of knowledge and awareness in the scientific community and among healthcare providers and administrators about aptamers and the possibilities they offer, as opposed to familiarity with antibodies.

From a technical perspective, more scientific investigations and improvements are needed along the developmental pipeline of aptamer-focused technologies to accelerate the translation of aptasensors into clinical settings. For example, robust ways for the identification of high-quality aptamers (e.g., SELEX-integrated microfluidics, artificial intelligence with high throughput sequencing, and appropriate affinity testing methods) are needed while reducing the duration of raising aptamers from the order months to only a few days or hours. In parallel, selection efforts must focus on clinically valuable targets; and downstream aptasensors must exhibit high stability and reproducibility and be adapted to test complex clinical samples so that a connection with “real life” applications in the clinic is created. In other words, proposed aptasensors become more than just proofs-of-concept. It is also important to take advantage of aptasensor multiplexing for high-throughput, low-cost sensing of multiple targets. The sensitivity and specificity of aptasensors can be enhanced by the incorporation of nanomaterials such as nanoparticles, quantum dots, and carbon nanotubes which are gaining great attention for their unique physicochemical properties and performance, as well as their use as alternatives to traditional fluorophores facing photobleaching issues [236].

The majority of aptasensors are made using DNA aptamers. This is in part because producing longer RNA and the need for reverse transcription are a few extra steps with the selection of RNA aptamers against a given target. RNA aptamers, however, are known to be conformationally much more flexible and suitable for binding to an expanded repertoire of targets. After immobilization, these RNA and DNA aptamers must retain their high qualities (affinity and specificity) and stay resistant to nucleases. For this, pre-, mid-, or post-SELEX aptamer modifications can be employed. One of the technical challenges during and after surface immobilization is due to the nature of nucleic acids being negatively charged. This may lead to undesirable non-specific interactions resulting from electrostatics between the aptamer probes and their immobilization surface during fabrication [51,237]. Therefore, preliminary studies must be performed to understand the details of interactions between surface, aptamer, and target in a complex sample matrix. Although ideal aptasensors are designed for simplicity and cost-effectiveness, they may easily require technical expertise and become expensive when mass-produced using precious metals.

Despite these and other stringent limitations on aptasensors, these devices will soon provide valuable and unique alternative approaches to antibody-based detection methods as described throughout. Quick pinpointing and resolution of major bottlenecks in the R&D pipeline of aptansensing technologies will propel these devices into the market to facilitate the realization of their clinical diagnosis capabilities. Interestingly, beyond aptamer-antibody comparisons, new methods are progressing to show how aptamers accompanied by monoclonal antibodies provide highly sensitive joint sensing methods [238,239].

## Figures and Tables

**Figure 1 molecules-28-03728-f001:**
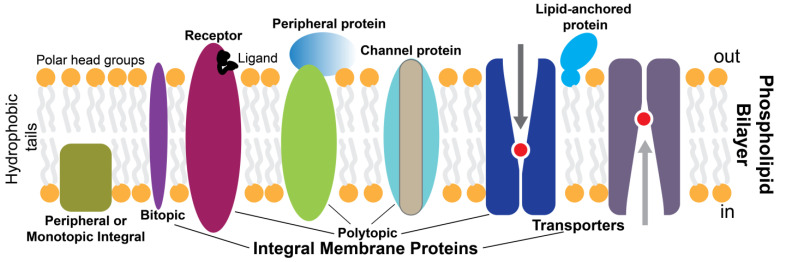
Simplistic overview of membrane proteins in a lipid bilayer.

**Figure 2 molecules-28-03728-f002:**
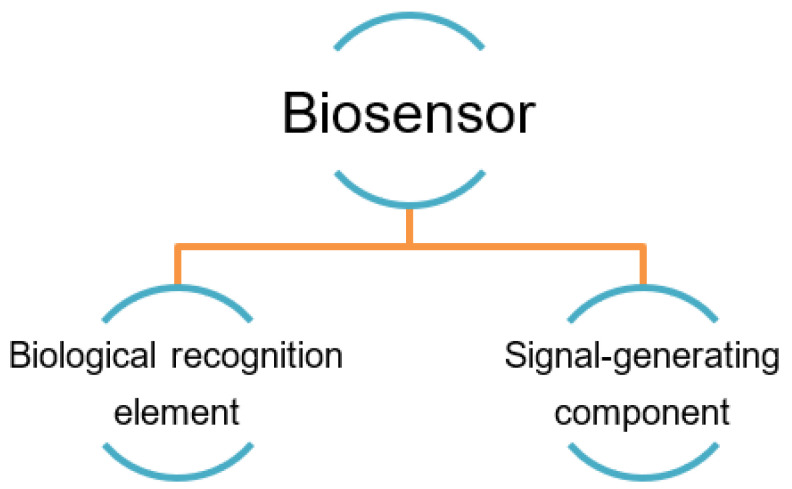
The two major generic parts of a biosensor [51].

**Figure 3 molecules-28-03728-f003:**
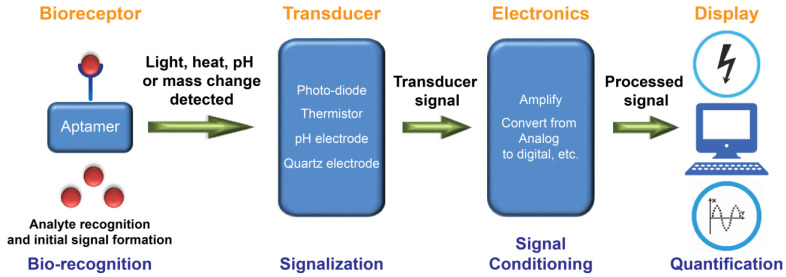
A schematic representation of the working principle of an aptamer-based biosensor (aptasensor). A bio-receptive aptamer recognizes the target with high affinity and specificity. The aptamer-target complex generates a signal in the form of light, heat, pH, mass change, etc. The generated biorecognition signal is detected, and a transducer commonly converts it into a measurable electrical or optical signal that is proportional to the number of interactions between the aptamer and its target. Electronics of a biosensor have complex circuitries that process signals arriving from the transducer (e.g., amplify and convert signals from analog to digital form). Finally, the display unit quantifies processed signals and allows users to interpret the data by displaying the output signal in numeric, graphic, tabular, or image forms. Adapted with permission from Ref. [45].

**Figure 4 molecules-28-03728-f004:**
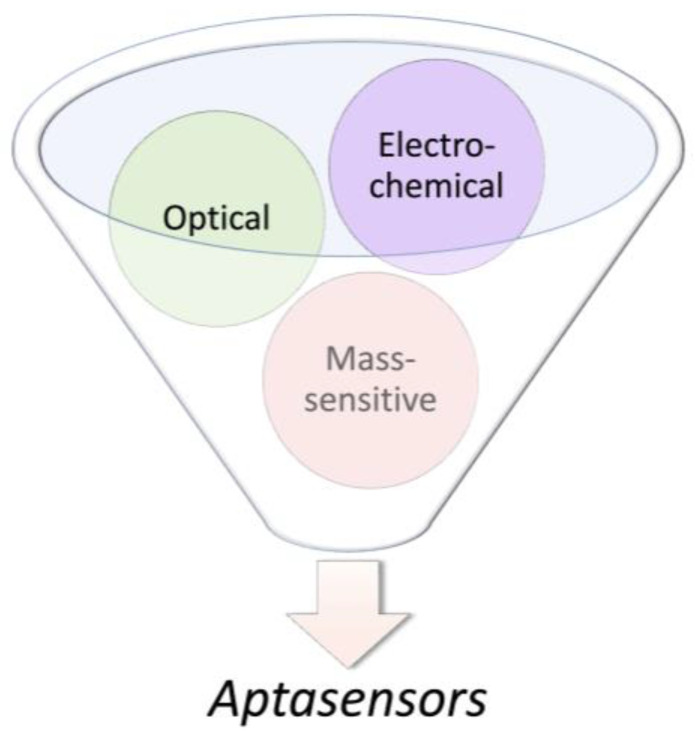
The most common divisions of aptasensors.

**Figure 5 molecules-28-03728-f005:**
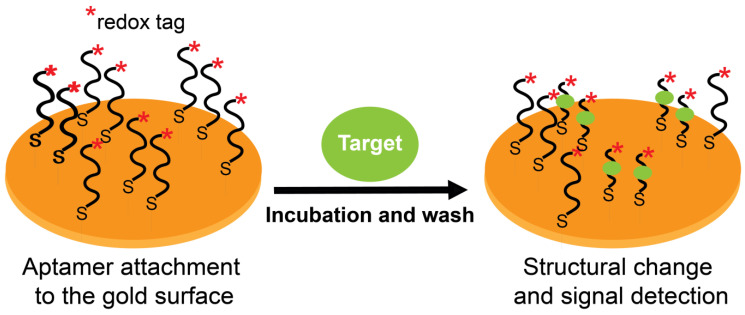
Working principle of redox-labeled electrochemical aptasensors. Target-specific aptamers are immobilized on a gold surface via thiol groups. Target binding induces a structural change (conformational flexibility and/or strand displacement) in the labeled aptamers, thereby facilitating (signal-on) or hindering (signal-off) electron transfer with the gold surface.

**Figure 6 molecules-28-03728-f006:**
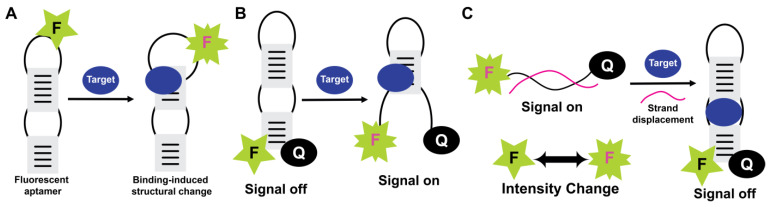
Fluorescence-based optical aptasensors. Fluorescence intensity-change as increase (**A**), (signal-on) (**B**), and decrease (signal-off) (**C**) formats are shown for fluorescently labeled aptamers. In the intensity-change format, fluorescence depends on target binding to promote conformational changes and subsequent structural reorganization. In “signal-on” and “signal-off” formats (i.e., aptabeacon-based designs), target-induced structural changes dictate liberation/quenching of the fluorescence signal (**F**: Fluorophore, **Q**: Quencher).

**Figure 7 molecules-28-03728-f007:**
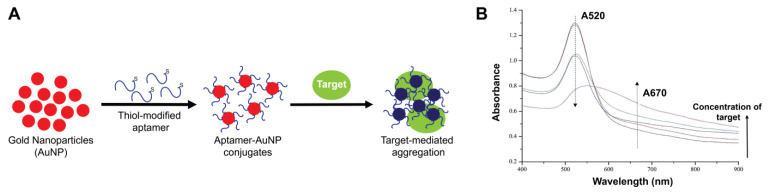
Working principle of a colorimetry-based optical aptasensor employing analyte-induced assembly of gold nanoparticles. Target binding to thiol-modified aptamers causes AuNPs aggregation and converts their color from red to blue (**A**). Absorbance levels (A670/A520) can be measured as a function of the concentration of the target in the medium (**B**).

**Figure 8 molecules-28-03728-f008:**
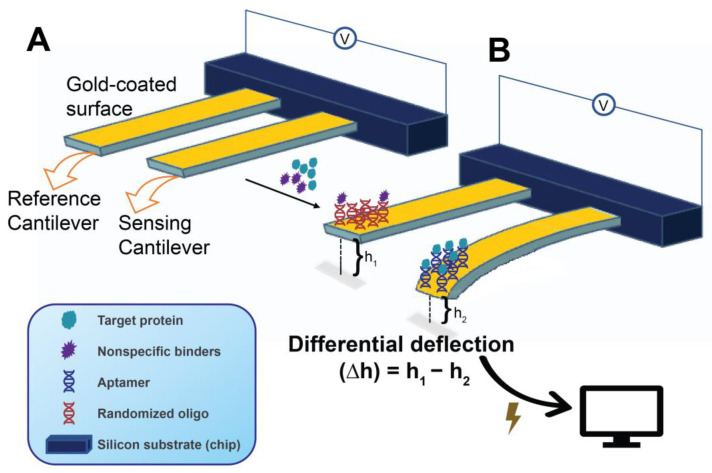
Working principle and main components of a microcantilever aptasensor in surface compressive stress mode. Reference and sensor platforms (usually in the same sample loading compartment) are shown prior to surface functionalizing (**A**) (i.e., naked and cannot bind and bend yet) by immobilization of suitable oligonucleotides. An operation aptasensor (**B**) after immobilization of a target-binding nucleic acid aptamer (sensory cantilever) and a randomized oligo (reference cantilever). The detected bending is transduced into measurable electrical signals (voltage/current changes) that allow the calculation of target protein concentration. The deflection caused by target binding is often reported as differential deflection (∆h = h_1_ − h_2_).

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
