# Peer review of "Aptamers Targeting Membrane Proteins for Sensor and Diagnostic Applications"

_molecules, 2023, doi:10.3390/molecules28093728_

Round 1
Reviewer 1 Report
The authors mainly focus on the aptamer-based MP-sensing technologies for diagnostic applications and have classified the aptasensors into several types based on their working techniques. This review is interesting and meaningful, and I recommend to you that minor revision is needed before the manuscript is published. Corresponding comments are as follows:
1. It is better to carefully check the format of the numbers in the manuscript, such as [Ru(NH3)6]3+, Fe3O4, 1.10 x 105 cfu/mL and so on.
2. The authors mention in Page 4 line 173 that the Kd ranges from pM to μM, while there are examples of Kd with mM level, such as the aptamer for glucose.
3. Page 9 line 240, the words “Their Value” should be removed from the title.
4. The conjugated polymer-based colorimetirc method is also a typical colorimetirc aptasensors which should be discussed in the part of colorimetry-based method. Corresponding references, such as Anal. Chem. 2022, 94, 44, 15456–15463 and ACS Sens. 2022, 7, 686−703, should be added.
Author Response
The authors would like to thank the reviewer for their invaluable suggestions. We believe the manuscript reads better with all the changes and additions.
- It is better to carefully check the format of the numbers in the manuscript, such as [Ru(NH3)6]3+, Fe3O4, 1.10 x 105 cfu/mL and so on.
Thanks for the reminder, we have reformatted numbers and chemical formulae according to Molecules specifications.
- The authors mention in Page 4 line 173 that the Kd ranges from pM to μM, while there are examples of Kd with mM level, such as the aptamer for glucose.
We have mentioned the term “generally” in that sentence as most of the aptamer Kds fall between pM and μM. We are aware of the aptamers developed for small molecules like amino acids and glucose that have Kds in mM range.
- Page 9 line 240, the words “Their Value” should be removed from the title.
Good point, we have removed “Their Value” from the text.
- The conjugated polymer-based colorimetirc method is also a typical colorimetirc aptasensors which should be discussed in the part of colorimetry-based method. Corresponding references, such as Anal. Chem.2022, 94, 44, 15456–15463 and ACS Sens. 2022, 7, 686−703, should be added.
References were incorporated into the text.
Reviewer 2 Report
The review is very nice and clear. I have one major suggestion and a few minor comments that can increase the value of the article
My major comment is about the choice of methods. One of very interesting area of optical sensors is SERS-based aptasensors. A dozen of nice works on SARS-CoV-2 and influenza A detection can be found, e.g. 10.3389/fbioe.2022.1076749, 10.1021/acssensors.1c00596, 10.3389/fchem.2022.937180
Minor comments are connected with the language and some terms:
line 113 reimbursable (commercially available or something else?)
line 116 … influenza viral cell surface glycoprotein (may be surface glycoprotein of influenza virus?)
line 118 ‘More recently, angiotensin-converting enzyme II (ACE2) has become an important diagnostic and therapeutic target in the fight against COVID-19’ [35]. ACE2 is a therapeutic target but not a matter for COVID-19 diagnostics
line 146 ‘Biological sensors, on the other hand, have become ubiquitous precursors…’ term ‘precursor’ is unclear
line 194 ‘Compared to antibodies, aptamers are highly malleable’. It is not true there are a lot of highly stable DNA aptamers, for example, G-quadruplex-based aptamers.
Figure 3 There are many examples with spectral changes in biosensor (fluorescence, Raman spectra etc.); they can’t be described as ‘light emission’.
The text contains several mistakes, e.g. line 365 ‘5.2.1. Flurosence-based’
Author Response
The authors would like to thank the reviewer for their invaluable suggestions. We believe the manuscript reads better with all the changes and additions.
The review is very nice and clear. I have one major suggestion and a few minor comments that can increase the value of the article.
My major comment is about the choice of methods. One of very interesting area of optical sensors is SERS-based aptasensors. A dozen of nice works on SARS-CoV-2 and influenza A detection can be found, e.g. 10.3389/fbioe.2022.1076749, 10.1021/acssensors.1c00596, 10.3389/fchem.2022.937180
We have now integrated “SERS-based aptasensor” in the review including the examples from the indicated references as well as one of our recently published studies.
Minor comments are connected with the language and some terms:
line 113 reimbursable (commercially available or something else?)
We have removed it from the text for clarity.
line 116 … influenza viral cell surface glycoprotein (may be surface glycoprotein of influenza virus?)
We would like to thank the reviewer as we have updated the text according to their suggestions.
line 118 ‘More recently, angiotensin-converting enzyme II (ACE2) has become an important diagnostic and therapeutic target in the fight against COVID-19’ [35]. ACE2 is a therapeutic target but not a matter for COVID-19 diagnostics
We agree with the reviewer, but instead of deleting the “diagnostic” from the text, we have revised that section to include other proteins used for diagnostics.
line 146 ‘Biological sensors, on the other hand, have become ubiquitous precursors…’ term ‘precursor’ is unclear
Now we have removed the term for “precursor” – and used “platforms”.
line 194 ‘Compared to antibodies, aptamers are highly malleable’. It is not true there are a lot of highly stable DNA aptamers, for example, G-quadruplex-based aptamers.
We had used the term “malleable” not to mean something negative but rather to indicate aptamer structures are flexible and dynamic and also their chemical modification potentials. However, we have revised the section to indicate the aptamers with stable structures.
Figure 3 There are many examples with spectral changes in biosensor (fluorescence, Raman spectra etc.); they can’t be described as ‘light emission’.
We now have examples of other aptasensors that are represented through figures 5 to 8.
The text contains several mistakes, e.g. line 365 ‘5.2.1. Flurosence-based’
We have spell checked the file and thus we hope that we eliminated such mistakes.
Reviewer 3 Report
The review by Kara et al provided a good summary in describing the current methods for aptamer-based biosensors and used typical results to highlight the past successes of this field. It is an interesting and up-to-date review. I have a few suggestions, mainly on the conceptual architecture of this review.
1) I found that this technical description is difficult to understand. It would be very helpful to include graphic descriptions of technical principles behind the key detection methods in Section 5. A table comparing pros and cons of these different methods in a concise way would be important for the readers to gain a big picture.
2) The MP classification presented in Fig. 1 does not include the half-entry proteins that only span one leaflet. The famous example is the caveolins. The other class are protein complexes that span two bilayer membranes, which can be present in different microbe surfaces and be useful for screening. There are also lipid-anchored proteins that are not presented either.
3) Even though the enthusiasm for the aptamer has been high, the review did mention that so far none of the aptamers have gone to the clinical trials nor to the market. It is important for the review to highlight the main technical problems or scientific limitations. Is it immune rejection, the delivery, the BBB, or incompatibility with the antibody-based methods? From these limitations, it would be good to propose new ways to overcome the main issues.
4). After the description of the different detection methods, it would be helpful if the review can raise an outlook on what is promising to make the aptamer-based methods easy to be adopted for research and translation, and help push at least some to the market in the near future. Otherwise, these will stay limited in being good entertaining research tools.
5). Pg2 : the RTKs can be separated from the receptors, instead of being fused to the intracellular part of a receptor.
6).line 240. “to MPs Their Value” does read right.
Author Response
The authors would like to thank the reviewer for their invaluable suggestions. We believe the manuscript reads better with all the changes and additions.
The review by Kara et al provided a good summary in describing the current methods for aptamer-based biosensors and used typical results to highlight the past successes of this field. It is an interesting and up-to-date review. I have a few suggestions, mainly on the conceptual architecture of this review.
1) I found that this technical description is difficult to understand. It would be very helpful to include graphic descriptions of technical principles behind the key detection methods in Section 5. A table comparing pros and cons of these different methods in a concise way would be important for the readers to gain a big picture.
This is a great suggestion, however, the variables in each system are composed of more than simply aptamers. Rather, variations in performance are not only dependent on the sensor system but also on what matrix is being used and others that we have pointed out to the readers. Instead of a single table, we have now several other examples of aptasensors in Figures 5 through 8.
2) The MP classification presented in Fig. 1 does not include the half-entry proteins that only span one leaflet. The famous example is the caveolins. The other class are protein complexes that span two bilayer membranes, which can be present in different microbe surfaces and be useful for screening. There are also lipid-anchored proteins that are not presented either.
We have now updated the figure for clarity.
3) Even though the enthusiasm for the aptamer has been high, the review did mention that so far none of the aptamers have gone to the clinical trials nor to the market. It is important for the review to highlight the main technical problems or scientific limitations. Is it immune rejection, the delivery, the BBB, or incompatibility with the antibody-based methods? From these limitations, it would be good to propose new ways to overcome the main issues.
We found this a very good suggestion that strengthened our review. We now have included a section with aptamers in clinical trials and market, as well as the problems when applied in vivo.
4). After the description of the different detection methods, it would be helpful if the review can raise an outlook on what is promising to make the aptamer-based methods easy to be adopted for research and translation, and help push at least some to the market in the near future. Otherwise, these will stay limited in being good entertaining research tools.
We also have included a section with aptamers in clinical trials and market, as well as the problems when applied in vivo. Critical steps during SELEX and the importance of matrix were emphasized for successful aptamer applications.
5). Pg2 : the RTKs can be separated from the receptors, instead of being fused to the intracellular part of a receptor.
We believe that there are RTKs with (intracellular) kinase domain and RTKs associated with kinases (e.g. Janus kinases). So, we have kept that section unchanged.
6).line 240. “to MPs Their Value” does read right.
This is already addressed – mentioned by one of the other reviewers.
Round 2
Reviewer 2 Report
Thank уou for уour work. Now it is readу for publication